# Limited Role of Endogenous Vasopressin/Copeptin in Stimulation of ACTH–Cortisol Secretion during Glucagon Stimulation Test in Humans

**DOI:** 10.3390/biomedicines10112857

**Published:** 2022-11-08

**Authors:** Katarzyna Malicka, Wojciech Horzelski, Andrzej Lewiński, Krzysztof C. Lewandowski

**Affiliations:** 1Department of Endocrinology and Metabolic Diseases, Polish Mother’s Memorial Hospital-Research Institute, 93-338 Lodz, Poland; 2Faculty of Mathematics and Computer Science, University of Lodz, 90-238 Lodz, Poland; 3Department of Endocrinology and Metabolic Diseases, Medical University of Lodz, 93-338 Lodz, Poland

**Keywords:** copeptin, glucagon stimulation test, growth hormone, pituitary, vasopressin

## Abstract

Copeptin is a stable part of a vasopressin precursor that closely mirrors arginine vasopressin (AVP) secretion. It is known that AVP/copeptin is also released in response to nonosmotic stimuli, such as stress evoked during anterior pituitary dynamic testing. In order to examine the role of AVP in challenging the hypothalamo-pituitary-adrenal axis, we assessed adrenocorticotropic hormone (ACTH), cortisol, copeptin and growth hormone (GH) during a glucagon stimulation test (GST) in 10 patients with satisfactory initial cortisol concentrations (mean ± SD: 20.34 ± 5.10 µg/dL) and failure to show any further cortisol increment on stimulation. For comparison, we measured copeptin in two subjects during an insulin tolerance test (ITT). During GST, there was an increase in copeptin (*p* = 0.02, average individual increase of 98%, range 10% to 321%). There was a robust increase in GH (*p* = 0.002, average increase 3300%), a decline in cortisol (*p* = 0.02, average decline 21.8%) and a fall in ACTH (*p* = 0.06). The relative increase in copeptin during ITT (176% and 52.2%) overlapped with increments observed during GST; however, here there was an increase in cortisol (20.45→24.26 µg/dL and 4.23→29.29 µg/dL, respectively). There was a moderate correlation between copeptin and GH concentrations (*r* = 0.4235, *p* = 0.0007). These results confirm that AVP is not crucial for ACTH–cortisol stimulation, though it might be an important factor in GH secretion.

## 1. Introduction

The precise mechanism responsible for anterior pituitary stimulation in classic dynamic tests, such as an insulin tolerance test (ITT) or a glucagon stimulation test (GST) still remains unclear. It has long been postulated that vasopressin (AVP) is also released in response to nonosmotic stimuli and comprises part of the individual stress reaction [1]. AVP measurements are, however, troublesome; therefore, they are rarely used in clinical practice. Lately, AVP measurements have been successfully replaced by copeptin, which is secreted in equimolar amounts with AVP, remains stable and can be measured by several assays [2,3,4]. Copeptin is a part of the AVP precursor and its release was described during an insulin tolerance test (ITT) [5,6], a glucagon stimulation test (GST) [7,8] and a corticotropin-releasing hormone (CRH) stimulation test [9]. Therefore, it was suggested that vasopressin may be one of the factors that significantly influence the adrenocorticotropic hormone (ACTH)—cortisol secretion during dynamic tests of pituitary function. GST is commonly used for assessment of the anterior pituitary reserve; in fact, being the most popular test for the combined assessment of cortisol and growth hormone (GH) secretion in the UK [10].

Though GST is associated with an increase of copeptin [7,8], it is not clear whether AVP/copeptin stimulates the secretion of ACTH and/or cortisol, or is just an epiphenomenon, where copeptin is secreted independently from ACTH and/or GH; i.e., without any significant impact on ACTH/cortisol or GH release. Furthermore, mechanisms for the secretion of GH and ACTH/cortisol during GST might be different as there are patients with an intact anterior pituitary who show brisk GH secretion, but fail to demonstrate any significant increment in cortisol concentrations. On the strength of that, we decided to analyse copeptin secretion in a selected group of patients who underwent GST. These patients were characterised by satisfactory initial cortisol concentrations; however, they failed to show any further cortisol increments above the baseline level.

Assuming that AVP does play a role in ACTH, cortisol and GH secretion, i.e., acting as a direct or indirect stimulant for ACTH, we expected copeptin concentrations to rise during GST together with an increase in ACTH/cortisol and/or GH. On the other hand, a significant increase in copeptin concentrations without any further increase in ACTH/cortisol or GH would imply that copeptin does not play a significant role in the stimulation of ACTH/cortisol or GH. Some healthy patients during GST demonstrate, however, an increase in GH without a concomitant further increase in ACTH/cortisol. If there is a significant increase in copeptin in such circumstances, this might imply that copeptin may potentially play a role in GH secretion, but is less important for the secretion of ACTH/cortisol.

## 2. Materials and Methods

The study involved 10 adult patients selected from a larger group of 80 patients who underwent GST while hospitalised in the Department of Endocrinology and Metabolic Diseases of the Polish Mother’s Memorial Hospital, Research Institute, Lodz, Poland. The inclusion criterion was defined as satisfactory initial cortisol concentration (>375 nmol/L, i.e., 13.6 µg/dL); thus, confirming hypothalamic-pituitary-adrenal (HPA) axis integrity [11] with no further cortisol increment during stimulation. The study group consisted of 9 females and 1 male, aged 41.4 ± 15.2; BMI 25.4 ± 5.8 kg/m^2^ (mean ± SD). Five of those patients had a history of pituitary adenoma; however, they required no hormonal treatment. The rest comprised a heterogenous group diagnosed with hyperandrogenism *n* = 2, irregular menses *n* = 2, and reactive hypoglycaemia *n* = 1. Glucagon (GlucaGen 1 mg HypoKit, Novo Nordisk, Bagsvaerd, Denmark) was administered intramuscularly after an overnight fast at the dose of 1 mg or 1.5 mg depending on the patient’s weight; i.e., 1.5 mg for those over 90 kg. ACTH, cortisol, copeptin and GH concentrations were assessed according to GST “short protocol”; i.e., at 0–60–90–120–150–180 min.

For comparison, copeptin was also measured during a standard insulin tolerance test (ITT) in two other male patients: one with a history of pituitary surgery for macroadenoma (age 33, BMI 23.2 kg/m^2^) and the other with a history of head trauma (age 29, BMI 28 kg/m^2^).

Plasma ACTH and serum GH concentrations were measured using the immunochemiluminescence assay IMMULITE 2000 Xpi (Siemens, Munich, Germany). Serum cortisol was measured by the means of electrochemiluminescence assay on Cobas 6000 platform (Roche, Basil, Switzerland). Serum copeptin concentrations were measured in one batch with an immunoluminometric assay (B∙R∙A∙H∙M∙S CT-proAVP, Thermo Fischer Scientific B∙R∙A∙H∙M∙S GmbH, Hennigsdorf, Germany), as described before [3].

The Ethics Committee of the Polish Mother’s Memorial Hospital, Research Institute approved the study protocol (decision no. 63/2020) and written informed consent was obtained from all the participants.

### Statistical Analysis

Statistical analysis was performed by means of MedCalc Software 12.6.1 (Ostend, Belgium). The results are expressed as means ± SD. The results obtained at the point of maximal stimulation were compared to baseline values using a paired samples Wilcoxon test and by ANOVA for multiple measurements. Correlation coefficients were calculated using Spearman’s rank correlation test. Statistical significance was assumed at *p* < 0.05.

## 3. Results

After glucagon administration, typical changes in blood glucose, i.e., an initial rise and subsequent fall, were observed. There was a borderline non-significant (*p* = 0.06) fall in ACTH concentrations. Initial cortisol concentrations were quite high (mean ± SD: 20.48 ± 4.9 µg/dL) and decreased during stimulation, reaching 12.5 ± 5.10 µg/dL at 150′ (*p* = 0.02). In contrast, there was a significant increase in copeptin concentrations in all the patients (from 4.35 ± 2.62 pmol/L to 6.93 ± 3.8 pmol/L 0 vs. 180 min, *p* = 0.02), accompanied by a highly significant several-fold increase in GH concentrations (e.g., from 0.87 ± 1,47 ng/mL to 6.42 ± 5.72 ng/mL, 0 vs. 150 min, *p* = 0.002 (Figure 1, Table 1)).

There was no significant correlation between copeptin and ACTH or cortisol; however, there was a moderate (*r* = 0.4235), but significant (*p* = 0.0007) correlation between the copeptin and GH concentrations.

Individual data on the variation of cortisol, GH and copeptin are presented in Table 2. As it is well known that maximal GH and cortisol stimulation during GST is typically observed at 150 or 180 min [12], we compared the initial values (0 min) to the values obtained at 120–180 min of GST. Relative increase was calculated. The average individual increase in copeptin concentration was 98% (from 10% to 321% in individual patients).

Effects of ACTH, cortisol and GH during ITT in two other patients are presented in Table 3. Both subjects had diagnostic ITT, as reflected by a fall in glucose below 2.2 mmol/L (40 mg/dL). The relative increase in copeptin concentrations during ITT (176% and 52.2%) overlapped with the individual increments observed during GST; however, in contrast to GST, there was an increase in cortisol secretion in comparison to the initial levels (from 20.45 µg/dL to 24.26 µg/dL and from 4.23 to 29.29 µg/dL, respectively). Similarly to the GST group, GH secretion was also significantly stimulated.

## 4. Discussion

An insulin tolerance test and a glucagon stimulation test have been used for decades as a method of assessing anterior pituitary function in terms of stress-induced ACTH/cortisol and GH secretion. It has been proven that hypoglycaemia, being a nonosmotic stimulus, leads to a three-fold increase in AVP concentration [1]. Both CRH and AVP have specific receptors on the corticotroph surface and cooperate in activation of the hypothalamic-pituitary-adrenal axis [13], though precise mechanisms still remain to be elucidated. Measurement of copeptin, which reflects AVP secretion, was performed during dynamic pituitary testing. Previously, we have demonstrated that in patients with a history of pituitary disease after CRH administration, the HPA axis may be effectively stimulated even if copeptin secretion is attenuated. In particular, patients after pituitary surgery (i.e., with a subtle subclinical pituitary defect), who had a perfectly normal ACTH–cortisol response to CRH, failed to show any significant copeptin increase after CRH [9]. This observation suggested that AVP and ACTH/cortisol are released after CRH in a manner being largely independent from each other, where CRH appears to be much stronger and a direct stimulus for ACTH secretion, while AVP/copeptin secretion appears to be rather a weak stimulus for ACTH release in comparison to CRH. Our latest results, described in this article, seem to confirm that AVP is not crucial for ACTH/cortisol secretion, as some patients during GST fail to respond with an ACTH/cortisol increment even if copeptin concentrations increase significantly during stimulation.

Our study has its limitations regarding a small number of patients and heterogeneity of the study group. It should be emphasized, however, that the patients were selected from a larger group of 80 patients who underwent GST in our Department, where in typical circumstances there is an increase in both GH and ACTH–cortisol after glucagon stimulation. We have chosen the subjects with an unequivocally functional ACTH–cortisol axis [11]. This condition would be fulfilled even if more strict morning cortisol cut-off (506 nmol/L, i.e., 18.3 µg/dL) for a Roche^®^ cortisol assay [14] were applied.

The choice of the morning cortisol cut-off of (>375 nmol/L, i.e., 13.6 µg/dL) as suggested by Yo et al. [11] may be the matter of debate as a much lower cortisol cut-off of 8.8 µg/dL has been also suggested for fixed-dose GST [15]. On the other hand, Steeten et al. [16] demonstrated a clear-cut secondary adrenal failure in a patient with a history of large-cell lymphoma, treated with whole brain irradiation and a cortisol concentration of 16.6 µg/dL (462 nmol/L) before a Synacthen test. We cannot, however, rule out a possibility of the so-called “ceiling effect”; i.e., that AVP/copeptin in the range of concentrations observed during GST fails to stimulate further cortisol secretion in the setting of relatively high initial cortisol concentrations. Indeed, in eight out of our ten subjects, we observed higher baseline cortisol concentrations than the mean stimulated cortisol levels (17.26 µg/dL, i.e., 481 nmol/L) during GST in the study by Littley et al. [17]. There is also a mild (about 2 µg/dL) though non-significant increase in cortisol concentrations during GST between 120 and 180 min of GST (Figure 1). On the other hand, during ITT, we observe a further increase in cortisol from a rather high initial value of 20.45 µg/dL to 24.26 µg/dL, with a 176% increase in copeptin. At the same time, a similar copeptin increase (156%) was not accompanied by any significant increase in cortisol above the baseline level in subject nr 8 (Table 1), with an initial cortisol concentration of 14.8 ug/dL during GST (in fact, there was a fall in cortisol to 10.94 ug/dL between 120-180 min of GST). That implies that a relatively high initial cortisol concentration does not preclude any further increase, provided a stimulus is strong enough (the highest cortisol concentration—56.5 µg/dL (1565 nmol/L)—that we had seen was in a healthy junior doctor who briefly fainted during the exam and had her cortisol measured soon after coming round. Our data point, however, to a conclusion that a relative increase in AVP/copeptin in the concentrations observed during GST is unlikely to provide a strong enough stimulus to cause an increase in cortisol above the baseline levels if the concentrations of cortisol are above 14.8 µg/dL (410 nmol/L); i.e., the lowest baseline cortisol concentration in our GST subjects. This does not preclude the possibility that a massive increase in AVP/copeptin, e.g., during hypovolaemic shock or myocardial infarction, might provide a stimulus for cortisol secretion.

The possibility of the “ceiling effect” is also likely when we compare our data to results of the recent elegant study by Atila et al. [8]. Namely, baseline copeptin concentrations in the subjects undergoing GST in our study (4.35 pmol/L) were very similar to the baseline copeptin concentrations in (4.38 pmol/L) in the study by Atila et al. [8]; however, in our study that included subjects with relatively high initial cortisol concentrations, an increase in copeptin was less robust (i.e., to a mean of 6.93 pmol/L at 180 min of GST versus 12.08 pmol/L in a study by Atila et al. [8] and 15.9 pmol/L in our previous study [7]). The notion of less robust copeptin secretion in response to GST in subjects with relatively high baseline cortisol concentrations might be in keeping with the “ceiling effect”, though on the other hand, a relatively modest increase in copeptin (52.2%) with a rather low initial concentration (2.3 pmol/L) was accompanied by a marked cortisol increase (from 4.23 µg/dL to 29.29 µg/dL) in our second ITT subject. Unfortunately, despite an excellent general design (placebo control, taking into account the effect of nausea during GST, etc.), in the study of Atila et al. [8], cortisol concentrations were measured only at 0 and 150 min of GST, while maximal cortisol concentrations are commonly observed at 180 min. Yet, analysis of the results of the study of Atila et al. [8] shows that the overall cortisol increase from the baseline during GST in healthy subjects was relatively modest (mean increase of only 45 nmol/L (1.62 µg/dL) with the range from minus 52 nmol/L up to 240 nmol/L). This indicates that there were subjects who either failed to demonstrate any cortisol increase above the baseline, or even demonstrated a fall in cortisol concentrations. Unfortunately, the data on the individual variation of copeptin concentrations in the subjects failing to show any cortisol increments above the baseline were not available for comparison to our results.

In such a setting, in our opinion, further research is needed in order to either confirm or to refute the “ceiling effect” hypothesis. We speculate that such a study might include analysis of the relative increases of copeptin and cortisol during GST in relation to the initial (baseline) cortisol levels in order to identify the cortisol concentration above which an increase in copeptin might not be associated with any further cortisol increments.

In our study, during both GST and ITT, we observed marked elevations in GH concentrations. This suggests that AVP/copeptin in the range of the concentrations observed during GST might be involved in GH secretion rather than the ACTH/cortisol release. Interestingly, the mechanisms responsible for ACTH–cortisol and GH secretion during GST appear to be different. The link between AVP and GH has been postulated before. It has been proven that AVP stimulates GH release in goats [18]. Arvat et al. [19] demonstrated that glucagon and hexarelin (a GH secretagogue) have a synergistic effect on GH release. On the other hand, the administration of somatostatin reduced the AVP response during ITT, suggesting that GH may also influence AVP secretion [20].

## 5. Conclusions

In summary, our study suggests a possible direct interaction between AVP/copeptin and GH release during GST. The effect of AVP on ACTH and cortisol secretion during GST seems to be less relevant, at least in subjects with relatively high baseline cortisol concentrations. There is also a possibility of the “ceiling effect”; i.e., that AVP/copeptin fails to stimulate any further cortisol increase above a certain baseline cortisol concentration. Further research is, however, necessary in order to either confirm or to refute this hypothesis. On the strength of that, the precise mechanism (or mechanisms) responsible for ACTH and cortisol secretion during the glucagon stimulation test still remain to be elucidated.

## Figures and Tables

**Figure 1 biomedicines-10-02857-f001:**
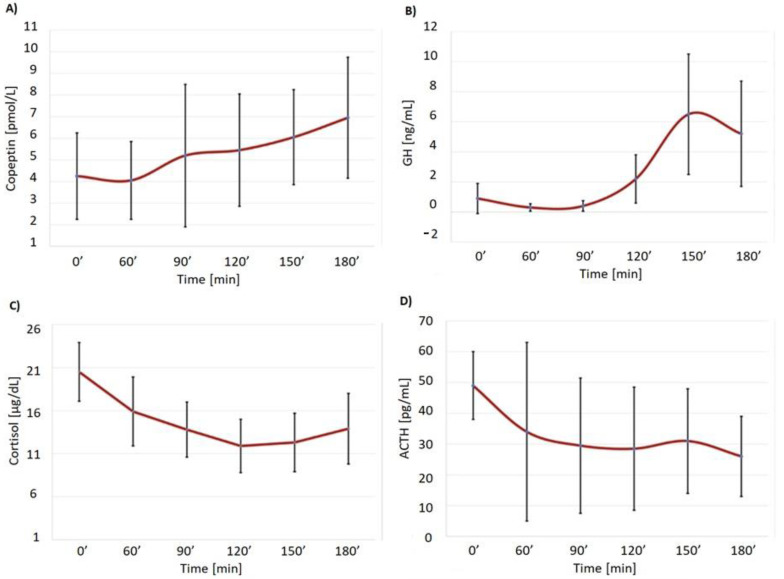
Copeptin (**A**), GH (**B**), cortisol (**C**) and ACTH (**D**) secretion during GST (means ± SD) in patients with satisfactory initial cortisol concentrations, but no further cortisol increment during stimulation.

**Table 1 biomedicines-10-02857-t001:** Copeptin (pmol/L), growth hormone (ng/mL), ACTH (pg/mL), cortisol (µg/dL) and glucose (mg/dL) secretion during GST (means ± SD) in patients with satisfactory initial cortisol concentrations, but no further cortisol increment during stimulation (ANOVA).

Time (min)	Copeptin (pmol/L) *		GH (ng/mL) **		ACTH (pg/mL) #		Cortisol (µg/dL) ***		Glucose (mg/dL)	
	Mean	SD	Mean	SD	Mean	SD	Mean	SD	Mean	SD
0	4.35	2.62	0.87	1.47	42.34	13.67	20.48	4.9	84.2	9.14
60	4.13	2.32	0.34	0.34	32.13	25.86	15.79	5.53	132.3	45.4
90	5.29	4.49	0.46	0.53	27.43	19.32	13.64	4.46	115.2	49.1
120	5.42	3.56	2.19	2.10	27.59	17.85	11.82	3.97	96.0	33.3
150	6.13	3.09	6.42	5.72	28.96	15.49	12.50	5.10	81.5	22.9
180	6.93	3.80	5.10	4.86	25.60	12.60	13.75	5.89	84.1	36.7

* denotes significant increase compared to 0 min (*p* = 0.0201); ** denotes significant increase compared to 0 min (*p* = 0.002); *** denotes significant decrease compared to 0 min (*p* = 0.002); # *p* = 0.06 in comparison to 0 min.

**Table 2 biomedicines-10-02857-t002:** Copeptin, cortisol and GH levels in patients during GST-basal and maximal values observed 120-180 min after glucagon administration plus concentration relative change (%).

Patient	Copeptin 0′ (pmol/L)	Copeptin Max (pmol/L)	Increase (%)	Cortisol 0′ (µg/dL)	Cortisol Max 120–180′ (µg/dL)	Increase (%)	GH 0′ (ng/mL)	GH Max 120–180′ (ng/mL)	Increase (%)
1	2.7	6.4	137	19.63	16.81	−14.4	0.15	7.44	4857
2	5.1	8.7	70.6	26.81	17.60	−34.4	3.05	9.21	202
3	2.0	2.2	10.0	18.15	9.36	−48.4	0.09	1.02	1033
4	2.8	3.3	17.6	18.60	7.09	−61.9	0.21	4.96	2261
5	2.9	4.2	44.8	14.81	14.86	0.3	0.21	7.27	3361
6	6.2	9.0	45.2	18.75	15.34	−18.2	0.12	7.18	5883
7	6.4	16.7	160.9	19.02	17.15	−9.8	0.49	5.60	1043
8	2.3	5.9	156.5	14.80	10.94	−26.1	4.16	20.60	395
9	10.2	11.8	15.7	25.00	17.17	−31.3	0.07	8.66	12,271
10	2.9	12.2	320.7	29.24	28.26	−3.4	0.13	2.37	1723

**Table 3 biomedicines-10-02857-t003:** Copeptin, cortisol and GH levels in the two patients during ITT-basal and maximal values observed 30–120 min after insulin administration plus concentration relative change (%).

Patient	1	2
Copeptin 0′ (pmol/L)	2.5	2.3
Copeptin max 30-120′ (pmol/L)	6.9	3.5
Increase (%)	176	52.2
Cortisol 0′ (µg/dL)	20.45	4.23
Cortisol max 30-120′ (µg/dL)	24.26	29.29
Increase (%)	18.6	592
GH 0′ (ng/mL)	5.15	0.05
GH max 30–120′ (ng/mL)	40.0	9.14
Increase (%)	677	18,180

## Data Availability

The datasets used and/or analyzed within the framework of this study are available from the corresponding author on reasonable request.

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
