# Peer review of "Limited Role of Endogenous Vasopressin/Copeptin in Stimulation of ACTH–Cortisol Secretion during Glucagon Stimulation Test in Humans"

_biomedicines, 2022, doi:10.3390/biomedicines10112857_

Round 1

Reviewer 1 Report

Malicka et al report a study investigating copeptin, ACTH, cortisol, and GH during glucagon stimulation test in 10 patients (1 men, 9 women). Additionally, they measured copeptin in 2 men during insulin tolerance test. They interpret their finding that GH and AVP interact during GST while the effect of AVP on ACTH/cortisol seems to be less relevant.

1)    The whole purpose and novelty of the finding is unclear to the reviewer. While the authors speculate on the interaction between cortisol/ACTH and endogenous vasopressin, their findings mainly allow speculations and could be interpreted differently. They cite their previous trial investigating the remaining 70 subjects with what they call appropriate cortisol response. In this previous manuscript (Lewandowski 2016), the authors have identified a close correlation between cortisol/ACTH and copeptin/vasopressin, which they cannot confirm in this study.

2)    Could the authors clarify on the assay used in this and their previous study? In their previous study, the authors cite a study by Morgenthaler 2006 which used a different assay than the one mentioned in the submitted paper. As currently evaluated by Sailer 2021, copeptin assays might not be 100% comparable.

3)    Additionally, the authors should clarify whether copeptin was used prospectively in both tests or from frozen samples. Did the samples undergo further freeze-thaw cycles? This might influence and distort results of copeptin, ACTH and cortisol.

4)    What was the reason of only measuring 10 subjects during GST and 2 during ITT? A larger collective would give more information on the interplay of ACTH/cortisol, GH and AVP. Additionally, comparing mainly women undergoing the GST and only men undergoing the ITT might distort the results.

5)    The authors miss to mention a recent publication by Atila et al. (EJE 2022) who also investigate copeptin following GST. Even though their main interest was the differential diagnosis of GST in patients with central diabetes insipidus, they also report data of GH and cortisol.

6)    Regarding their interpretation that copeptin/vasopressin might not be the main stimulus for cortisol/copeptin, looking at the figures, it seems that there is a rise in cortisol at 150/180 corresponding to a continues rise in copeptin/vasopressin and a drop in ACTH. This last rise in cortisol could potentially be due to copeptin/vasopressin, leading to the interpretation that even though not primarily, copeptin/vasopressin may lead to increased cortisol if other systems fail to stimulate cortisol. Additionally, a copeptin below 10pmol/l is still somewhat in the normal range. In highly stressed patients, e.g., stroke, myocardial infarction, pneumonia, we expect copeptin levels to be around 50 if not higher. These levels might be necessary to stimulate cortisol. This would also be more plausible as an increased cortisol response to high copeptin levels might only be necessary in severe stress.

Author Response

1)    The whole purpose and novelty of the finding is unclear to the reviewer. While the authors speculate on the interaction between cortisol/ACTH and endogenous vasopressin, their findings mainly allow speculations and could be interpreted differently. They cite their previous trial investigating the remaining 70 subjects with what they call appropriate cortisol response. In this previous manuscript (Lewandowski, 2016), the authors have identified a close correlation between cortisol/ACTH and copeptin/vasopressin, which they cannot confirm in this study.

We thank the Reviewer for all useful comments that helped us to improve our paper.

We agree that the purpose of the study, i.e. investigation of the role of AVP/copeptin in stimulation of ACTH and/or GH release was stated not entirely clear, so we rephrased parts of the Introduction. It should be noted, however, that in our previous paper (Lewandowski et al. Copeptin under glucagon stimulation, 2006) we observed a correlation between ACTH/ and copeptin (strongest at 120 minutes and 150 minutes, in healthy controls, r=0.923, and r=0.86), but correlation between cortisol and copeptin  was not significant (Table 2). It is, however, recognized that apart from circadian, there is also an ultradian variation in ACTH and cortisol, so there is a time-shift between the pulse of ACTH and the pulse of cortisol.

2)    Could the authors clarify on the assay used in this and their previous study? In their previous study, the authors cite a study by Morgenthaler 2006 which used a different assay than the one mentioned in the submitted paper. As currently evaluated by Sailer 2021, copeptin assays might not be 100% comparable.

We are aware of differences between assays, so we quote the paper by Sailer et al. (now reference 4). In the studies we use the modified assay originally described by N.G. Morgenthaler, J. Struck, C. Alonso, A. Bergmann, Assay for the measurement of copeptin, a stable peptide derived from the precursor of vasopressin. Clin. Chem. 2006, 52, 112–119, with subsequent modification (Fenske et al. J Clin Endocrinol Metab 2011, 96, 1506-1515).

3)    Additionally, the authors should clarify whether copeptin was used prospectively in both tests or from frozen samples. Did the samples undergo further freeze-thaw cycles? This might influence and distort results of copeptin, ACTH and cortisol.

ACTH and cortisol were measured on the day of collection. Copeptin was measured from frozen samples (-80°C), however, just after one freeze-thaw cycle, i.e. all samples were handled in the same manner. It is well known that copeptin remains stable in such circumstances (Heida et al., Clin Chem Lab Med. 2017, 55, 984-992.doi: 10.1515/cclm-2016-0559).

4)    What was the reason of only measuring 10 subjects during GST and 2 during ITT? A larger collective would give more information on the interplay of ACTH/cortisol, GH and AVP. Additionally, comparing mainly women undergoing the GST and only men undergoing the ITT might distort the results.

We agree that larger group of investigated patients might be useful, but the number of subjects and their sex ratio was derived from an overall much larger number of investigated patients. The great majority of patients show an initial fall and then a marked increase in cortisol typically exceeding the baseline value, while lack of stimulation of further cortisol secretion occurs in about one in six subjects with an intact anterior pituitary function. Obviously in subjects who show both an increase in copeptin and cortisol it is not possible to differentiate whether copeptin is indeed a stimulus for ACTH-cortisol release due a simultaneous nature of these phenomena. On the other hand, a clear increase of copeptin in the setting of a flat, or even declining cortisol concentrations indicates that stimulatory effect of AVP/copeptin during GST is minimal, even if assume some minor effect reflected in a about 2 ug/dl (non-significant) increase in cortisol between 120 and 180 minutes of GST. We now comment extensively of on these issues in the discussion.

5)    The authors miss to mention a recent publication by Atila et al. (EJE 2022) who also investigate copeptin following GST. Even though their main interest was the differential diagnosis of GST in patients with central diabetes insipidus, they also report data of GH and cortisol.

We now quote this paper (reference 8) both in the Introduction and in the Discussion and compare their findings to our data.

6)    Regarding their interpretation that copeptin/vasopressin might not be the main stimulus for cortisol/copeptin, looking at the figures, it seems that there is a rise in cortisol at 150/180 corresponding to a continues rise in copeptin/vasopressin and a drop in ACTH. This last rise in cortisol could potentially be due to copeptin/vasopressin, leading to the interpretation that even though not primarily, copeptin/vasopressin may lead to increased cortisol if other systems fail to stimulate cortisol. Additionally, a copeptin below 10 pmol/l is still somewhat in the normal range. In highly stressed patients, e.g., stroke, myocardial infarction, pneumonia, we expect copeptin levels to be around 50 if not higher. These levels might be necessary to stimulate cortisol. This would also be more plausible as an increased cortisol response to high copeptin levels might only be necessary in severe stress.

We partially discuss this in our reply to question 4. Indeed, an increases of copeptin during GST are significant, but not as robust as in conditions associated during severe stress. Our study does not preclude a possibility that marked increase in AVP/copeptin, for instance during myocardial infarction, hypovolaemic shock, etc. might result in a significant secretion in cortisol. On the other hand our data suggest that more modest copeptin increases during GST probably play only a minor role (if any) in cortisol secretion. We now comment on this in the Discussion taking into account the comments off Reviewer 2 on possible “ceiling effect”.

Reviewer 2 Report

Summary:

This work by Malicka et al. reports interesting data regarding the influence of the AVP/copeptin on the ACTH/cortisol axis by retrospectively analyzing GST results of a sub-set of patients clinically evaluated for endocrine dysfunction. The authors conclude from this data that AVP is not a direct mediator of the ACTH/cortisol axis. Although this work adds novel findings to field of AVP/copeptin, particularly in regard to it’s role in non-osmotic stress, the elevated baseline cortisol levels in the subjects limits the conclusions that can be drawn from the study, thus dampening my overall enthusiasm for this manuscript.

Major issues:

1)     Given that the study does not measure AVP directly, the title as currently written is misleading.  At the very least it should be modified to read, “Limited Role of Endogenous Vasopressin/Copeptin in Stimulation . . .”

2)     The manuscript would be significantly improved by adding a more formalized hypothesis in the last paragraph of the introduction.  I had to read this section several times to understand that the authors were testing whether increased levels of copeptin can occur without concurrent increases in ACTH and cortisol.

3)     I do not agree  that (baseline) morning cortisol levels in isolation are entirely sufficient to establish normal functioning of the ACTH/cortisol axis, which is crucial for the authors conclusion that copeptin/AVP does not drive increases in ACTH/cortisol.  Although the authors cite a study that found that morning cortisol levels are predictive of adrenal sufficiency via SST, the unusually high AM levels in the study participants actually suggest the opposite, i.e. Overstimulation of the  ACTH/cortisol axis at baseline.  An alternative explanation of the data is that this high baseline ACTH/cortisol caused a ceiling effect whereby the observed increases in AVP/copeptin levels could not further stimulate ACTH/cortisol levels.  In fact, the baseline (pre-stimulation) cortisol levels of all subjects were already above the suggested lower cut-off pt for diagnosis of adrenal insufficiency via fixed dose- GST (Hamrahian et al. Pituitary 19, 332–341 (2016). Furthermore, 8 of 10 subjects included in the study had baseline cortisol levels that were higher than the mean “stimulated” cortisol levels reported in health subjects during a GST (Littley et al. Clin Endocrinol. 1989 Nov;31(5):527-33). Thus, the possibility of a ceiling effect needs to be addressed with further data from subjects with “normal” baseline cortisol levels.

4)     It is not entirely clear what the ITT data adds to this study.  It is my understanding that the authors include this data to demonstrate that “normal” copeptin, cortisol, and GH levels during hypoglycemia.  This is really unnecessary because the copeptin levels observed during the GST are entirely consistent with published studies that measured copeptin during GSTs.  The authors should either better justify the inclusion of this data or remove it.

Minor issues:

1)     Several abbreviations, such as AVP ACTH, are not defined at first use.

2)     The manuscript should be edited for language/clarity.  Several sentences are not clear as written, e.g. at LN 134-137.

3)     The quality of figure 1 is not consistent with the authors previous works, such as Lewandowski, LewiÅ„ski, et al. Endocrine (2017) 57:474–480 and Lewandowski, LewiÅ„ski, et al. Endocrine (2016) 52:344–351

Author Response

Major issues:

1)     Given that the study does not measure AVP directly, the title as currently written is misleading.  At the very least it should be modified to read, “Limited Role of Endogenous Vasopressin/Copeptin in Stimulation . . .”

 We have changed the title accordingly.

2)     The manuscript would be significantly improved by adding a more formalized hypothesis in the last paragraph of the introduction.  I had to read this section several times to understand that the authors were testing whether increased levels of copeptin can occur without concurrent increases in ACTH and cortisol.

We fully agree that our original hypothesis was not clearly presented. We have therefore  rephrased the Introduction accordingly.

3)     I do not agree  that (baseline) morning cortisol levels in isolation are entirely sufficient to establish normal functioning of the ACTH/cortisol axis, which is crucial for the authors conclusion that copeptin/AVP does not drive increases in ACTH/cortisol.  Although the authors cite a study that found that morning cortisol levels are predictive of adrenal sufficiency via SST, the unusually high AM levels in the study participants actually suggest the opposite, i.e. Overstimulation of the  ACTH/cortisol axis at baseline.  An alternative explanation of the data is that this high baseline ACTH/cortisol caused a ceiling effect whereby the observed increases in AVP/copeptin levels could not further stimulate ACTH/cortisol levels.  In fact, the baseline (pre-stimulation) cortisol levels of all subjects were already above the suggested lower cut-off pt for diagnosis of adrenal insufficiency via fixed dose- GST (Hamrahian et al. Pituitary 19, 332–341 (2016). Furthermore, 8 of 10 subjects included in the study had baseline cortisol levels that were higher than the mean “stimulated” cortisol levels reported in health subjects during a GST (Littley et al. Clin Endocrinol. 1989 Nov;31(5):527-33). Thus, the possibility of a ceiling effect needs to be addressed with further data from subjects with “normal” baseline cortisol levels.

We thank the Reviewer for these very useful remarks.

Indeed, the Reviewer has raised a very interesting and pertinent issue of the impact of further AVP/Copeptin release on cortisol concentrations in the setting of high initial cortisol levels. There is indeed a possibility that AVP/copeptin might fail to stimulate any further cortisol secretion during GST above certain baseline cortisol concentration, that may be described as a “ceiling effect”.  We have therefore amended the discussion to address this issue.

It must be recognized, however, that there are no universally recognised cut-off point for morning cortisol in diagnosing secondary adrenal insufficiency, therefore virtually any chosen cut-off point might be a matter of debate of either being too high or too low.  In such context a 8.8 ug/dl cortisol cut-off point, as suggested by Hamrahian et al. may be a matter of questionable, taking into account the previous data on baseline cortisol concentrations in subjects in the study of Streeten DH et al (J Clin Endocrinol Metab. 1996, 81, 285-290.doi: 10.1210/jcem.81.1.8550765) with confirmed secondary adrenal insufficiency. We also agree that our baseline cortisol concentrations were higher that mean stimulated cortisol levels after GST as reported by Littley MD et al. Clin Endocrinol (Oxf). 1989, 31, 527-533, i.e. 4181 nmol/l (17.26 ug/dl). In such setting a possibility of the “ceiling effects” is indeed likely, so we discuss this comparing our data to the recent study by Atila C (Eur J Endocrinol 2022 ref 8).

4)     It is not entirely clear what the ITT data adds to this study.  It is my understanding that the authors include this data to demonstrate that “normal” copeptin, cortisol, and GH levels during hypoglycemia.  This is really unnecessary because the copeptin levels observed during the GST are entirely consistent with published studies that measured copeptin during GSTs.  The authors should either better justify the inclusion of this data or remove it.

We comment on this in the discussion in the context of possible “ceiling effect” hypothesis, so we have decided to keep these data. 

Minor issues:

1)     Several abbreviations, such as AVP ACTH, are not defined at first use.

 This has been corrected

2)     The manuscript should be edited for language/clarity.  Several sentences are not clear as written, e.g. at LN 134-137.

 We have edited the manuscript to make it “more readable”.

 3)     The quality of figure 1 is not consistent with the authors previous works, such as Lewandowski, LewiÅ„ski, et al. Endocrine (2017) 57:474–480 and Lewandowski, LewiÅ„ski, et al. Endocrine (2016) 52:344–351

We have improved the quality of the figure.

Round 2

Reviewer 1 Report

The authors answered all questions satisfactory

Reviewer 2 Report

The authors have addressed my concerns.